# Optimal management of peripancreatic fluid collection with postoperative pancreatic fistula after distal pancreatectomy: Significance of computed tomography values for predicting fluid infection

Koki Maeda [1,2], Naohisa Kuriyama [1]*, Yuki Nakagawa[1], Takahiro Ito[1], Aoi Hayasaki[1], Kazuyuki Gyoten[1], Takehiro Fujii [1], Yusuke Iizawa[1], Yasuhiro Murata[1], Akihiro Tanemura[1], Masashi Kishiwada[1], Hiroyuki Sakurai[1], Shugo Mizuno[1]

1 Department of Hepatobiliary Pancreatic and Transplant Surgery, Mie University Graduate School of Medicine, Tsu, Mie, Japan, 2 Regional Medical Support Center, Mie University Hospital, Tsu, Mie, Japan

* naokun@med.mie-u.ac.jp

**Data Availability Statement:** All relevant data are within the paper.

## Abstract

Peripancreatic fluid collections have been observed in most patients with postoperative pancreatic fistula after distal pancreatectomy; however, optimal management remains unclear. This study aimed to evaluate the management and outcomes of patients with postoperative pancreatic fistula and verify the significance of computed tomography values for predicting peripancreatic fluid infections after distal pancreatectomy. We retrospectively investigated 259 consecutive patients who underwent distal pancreatectomy. Grade B postoperative pancreatic fistula patients were divided into two subgroups (B-antibiotics group and B-intervention group) and outcomes were compared. Predictive factor analysis of peripancreatic fluid infection was performed. Clinically relevant postoperative pancreatic fistulas developed in 88 (34.0%) patients. The duration of hospitalization was significantly longer in the B-intervention (n = 54) group than in the B-antibiotics group (n = 31; 41 vs. 17 days, p < 0.001). Computed tomography values of the infected peripancreatic fluid collections were significantly higher than those of the non-infected peripancreatic fluid collections (26.3 vs. 16.1 Hounsfield units, respectively; p < 0.001). The outcomes of the patients with grade B postoperative pancreatic fistulas who received therapeutic antibiotics only were considerably better than those who underwent interventions. Computed tomography values may be useful in predicting peripancreatic fluid collection infection after distal pancreatectomy.

## Introduction

Postoperative pancreatic fistula (POPF) remains the most critical complication of distal pancreatectomy (DP) [1]. Although several surgical innovations to prevent POPFs have been developed, the incidence of POPF after DP is still high at 11.0–49.1% [2–7]. Peripancreatic

**Funding:** The authors received no specific funding for this work.

**Competing interests:** The authors have declared that no competing interests exist.

fluid collections (PFCs) around the pancreatic stump were observed in most patients after DP [8]. Several studies revealed that PFCs even when large, often resolved easily, and interventional therapy was rarely required, even in patients with POPFs [9–11]. Meanwhile, Nappo et al. [12] showed that the frequency of radiological management after DP, such as percutaneous drainage and/or embolization, was higher than that after pancreaticoduodenectomy (PD). The management of PFCs, particularly in cases involving POPF, may differ among institutions or surgeons, and there is no consensus regarding the most appropriate treatment strategies. The optimal management of PFCs with POPF after DP therefore remains unclear. We divided the management of PFCs with POPF into conservative or invasive treatment strategies based on the patient's clinical burden. Conservative treatment usually requires therapeutic antibiotics, whereas invasive therapy requires interventions such as drainage. If the optimal indications for drainage of PFCs are clarified, it could be instructive for surgeons and potentially enhance the management of patients. Several studies have reported that microbial growth in POPFs is strongly associated with poor outcomes following pancreatic surgery [13, 14]; therefore, we focused on infection of PFC. We theorized that infected PFCs usually need to be drained. However, some non-infected PFCs might be improved by conservative therapy alone, without any drainage. If the development of infected PFCs could be predicted, it could aid in optimizing the selection of patients with PFCs who require interventions. Recently, the usefulness of computed tomography (CT) values in predicting infected abdominal fluid collections was reported [15, 16]. However, the usefulness of the CT values of PFCs after DP has not been reported. Therefore, the present study aimed to evaluate the management and outcomes of patients with grade B POPF after DP, and to ascertain the significance of CT values in predicting PFC infections to enhance POPF management.

## Methods

### Patients

Between January 2006 and December 2020, 259 consecutive patients who underwent DP at the Department of Hepatobiliary Pancreatic and Transplant Surgery of Mie University Hospital were retrospectively investigated. Patient demographics, preoperative characteristics, intraoperative details, and postoperative outcomes were retrospectively collected and analyzed. The protocol for this research was approved by a suitably constituted Ethics Committee at the institution (Committee of the Institutional Review Board at Mie University of Japan, Approval No. H2021-024), and the study conformed to the provisions of the Declaration of Helsinki. Prior to their inclusion in this retrospective study, individual participants provided informed consent for participation and the use of their medical records through an opt-out form. This consent procedure was approved by the ethics committee. All data were fully anonymized before we accessed them.

### Surgical technique

All surgeries were performed by or under the supervision of a board-certified expert hepatobiliary-pancreatic surgeon [17]. Both open and laparoscopic DP were performed using a unified surgical technique that involved transecting the pancreatic parenchyma and occluding its cut end using the hand-sewn or stapled closure technique. In the case of hand-sewn occlusion of the pancreatic cut end, the pancreatic parenchyma was transected using an ultrasonic coagulating dissector (SonoSurg; Olympus Optical Co. Ltd., Tokyo, Japan). The main pancreatic duct was ligated, and the cut end of the pancreatic parenchyma was occluded using the interrupted hand-sewn technique. Regarding the stapled closure technique, the pancreatic parenchyma was divided with a bare or mesh-reinforced triple-row stapler (NEOVEIL Endo GIA$^{TM}$

Reinforced Reload with Tri-Staple™ Technology 60 mm, COVIDIEN, North Haven, CT, USA) using a purple or black cartridge. This was based on the surgeon's judgment regarding the thickness at the pancreatic transection line. A closed suction drain was then placed in the peripancreatic and/or left subphrenic space.

## Postoperative management

For all patients, the administration of prophylactic antibiotics was routinely continued through postoperative day (POD) 2. Prophylactic somatostatin analogues were not used. The amylase content of the discharge from the closed-suction drain was evaluated at POD 1, 3, and until drain removal. In the absence of high amylase values > 3 times the ULN (upper limit of normal), drains were removed after POD 3. Bacterial cultures of the drain discharge were not routinely performed. Blood examination was routinely conducted preoperatively and at POD 1, 3, 6, or 7, and until discharge. Postoperative CT was routinely performed on POD 6–8 to evaluate PFCs.

Our basic therapeutic strategies for POPF after DP are summarized in Fig 1. A therapeutic antibiotic was immediately initiated when patients showed signs of clinical infection, such as a high-grade temperature ≥38.5°C and/or high inflammatory responses during blood examinations. Additional postoperative CT was performed to evaluate the source of infection when antibiotics were not clinically effective. Postoperative drainage was performed when a PFC was detected on postoperative CT and considered to be the source of inflammation. Ultrasound (US)-guided drainage was conducted by surgeons, but CT-guided drainage and transcatheter arterial embolization (TAE) procedures were performed by a radiologist. Endoscopic ultrasound (EUS)-guided drainage, endoscopic pancreatic drainage (EPD), and endoscopic hemostasis were performed by endoscopists who were also experts in the pancreatic field. Bacterial culture of the PFC was performed on the day of drainage.

## Definition and CT evaluation of peripancreatic fluid collections

A PFC was defined as a lesion >10 mm in diameter with a typical cyst-like appearance located at the pancreatic resection margin [9]. The CT value (in Hounsfield units [HU]) of the PFC was calculated by determining the median value of five thin slices of homogenous fluid collections in a region of interest while avoiding partial volume effects from the wall. The volume of the PFC was also measured using CT volumetry.

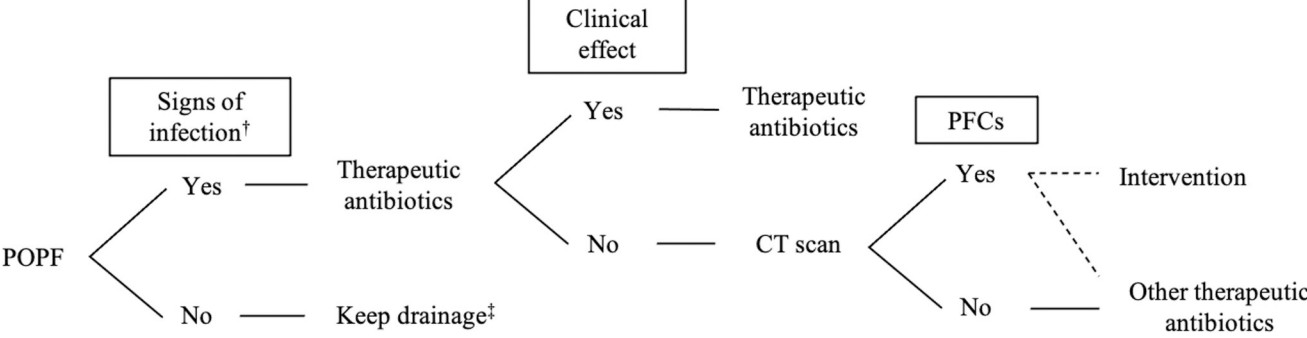

**Fig 1. Our basic therapeutic strategies for postoperative pancreatic fistula after distal pancreatectomy.** †High-grade temperature ≥ 38.5°C and/or high inflammatory responses. ‡Intraoperative drain is kept and exchanged once a week until the volume of discharge decreases. CT: computer tomography, DP: distal pancreatectomy, POPF: postoperative pancreatic fistula, PFCs: peripancreatic fluid collections.

## Subclassification of grade B POPF

The incidence and grade of POPF severity were determined according to the 2016 International Study Group for Pancreatic Surgery (ISGPS) classification [18]. We also defined POPF as any measurable volume of drain fluid during the postoperative course with an amylase level > 3 times the ULN. Postoperative complications were evaluated using the Clavien–Dindo classification of surgical complications and stratified as grades I–V [19, 20]. The diagnosis of the infected fluid collection was determined by a positive bacterial culture obtained from the PFC. We subclassified grade B POPFs into two subgroups based on the postoperative management (Fig 2). 1) Patients in the B-antibiotics group received therapeutic antibiotics only for the treatment of grade B POPF, while 2) patients in the B-intervention group received intervention without general anesthesia for the treatment of grade B POPF. Intervention included persistent drainage for >3 weeks and any non-surgical interventional procedures for PFCs with or without pharmacologic agents. In addition, the B-intervention group was further divided into two subgroups based on the PCF bacterial culture reports for predicting infection: B-intervention non-infected (patients without an infected PFC) and B-intervention infected (patients with an infected PFC). The patients who underwent interventions without PFC drainage were excluded from this classification. Patients who underwent EUS-guided drainage were also excluded from analysis due to the risk of bacterial contamination during the transgastric approach.

## Statistical analysis

Data are expressed as mean and range. The statistical significance of the continuous variables was tested using the Student's t-test or the Mann–Whitney U-test based on whether the data

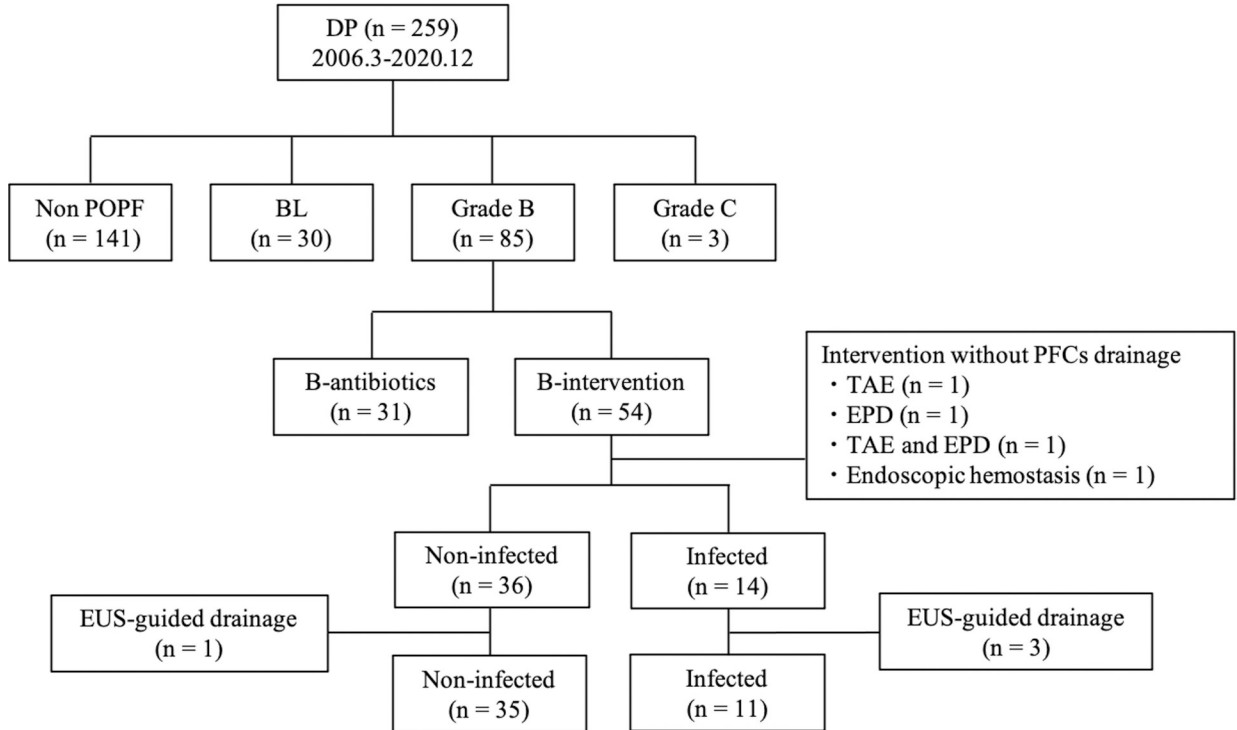

**Fig 2. Flowchart of the subclassification of grade B POPF.** DP: distal pancreatectomy, POPF: postoperative pancreatic fistula, BL: biochemical leak, PFCs: peripancreatic fluid collections, TAE: transcatheter arterial embolization, EPD: endoscopic pancreatic drainage, EUS: endoscopic ultrasonography.

were normally or non-normally distributed. The qualitative $\chi^2$ test was used for samples of nominal scales to compare the two groups. Variables with p < 0.05 after univariate analysis were entered into multivariate logistic regression analysis using the forward stepwise method. Comparisons were considered statistically significant at p < 0.05. The optimal CT cut-off values of the PFC were calculated using receiver operating characteristic (ROC) curve analysis using the Youden index. All statistical analyses were performed using IBM SPSS statistics version 26 software (IBM Japan, Tokyo, Japan) for Macintosh.

## Results

### Incidence and management of POPF after distal pancreatectomy

Patient demographics and clinical characteristics are summarized in Table 1. Of the 259 patients who underwent DP, 30 (11.6%) developed biochemical leaks (BL), and 88 (34.0%) developed clinically relevant (CR)-POPFs, classified as grade B in 85 patients (32.8%) and grade C in three patients (1.2%). A total of 70 patients (27.0%) developed ≧CDC grade IIIa complications. In this study, one patient without a POPF died from an unknown cause on POD 25. The incidence of PFCs at the first CT after DP was 79.9%.

As shown in Fig 2, patients were subclassified into B-antibiotic (n = 31) and B-intervention (n = 54) groups. All patients in the B-intervention group initially received antibiotics therapeutically, followed by the interventions. The duration and regimens of antibiotic therapy in both groups are shown in S1 Fig. Among the 54 patients in the B-intervention group, seven were treated with persistent drainage for >3 weeks, and 43 patients underwent additional drainage for PFCs. Among the patients with additional drainage, 34 underwent CT-guided drainage,

**Table 1. Characteristics of patients (n = 259) who underwent distal pancreatectomy.**

| Characteristics | Value |
|---|---|
| Sex, male/female | 151 / 108 |
| Age, years | 68 (3–89) |
| BMI, kg/m$^2$ | 21.4 (13.6–34.7) |
| Preoperative DM, n | 66 (25.5%) |
| Tumor characters, n | |
| Pancreatic ductal adenocarcinoma | 107 (41.3%) |
| Intraductal papillary mucinous neoplasm | 52 (20.1%) |
| Neuroendocrine tumor | 32 (12.4%) |
| Mucinous cystic neoplasm | 9 (3.5%) |
| Metastatic tumor | 9 (3.5%) |
| Others | 50 (19.3%) |
| Preoperative therapy, n | 68 (26.3%) |
| Intraoperative characteristics | |
| Operation time, min | 320 (132–830) |
| Blood loss, mL | 408 (0–11 300) |
| Laparoscopic surgery, n | 88 (34.0%) |
| Without splenectomy, n | 26 (10.0%) |
| Combined PV resection, n | 10 (3.9%) |
| Combined CA resection, n | 10 (3.9%) |
| Simultaneous resection of alimentary tract, n | 27 (10.4%) |

Data are expressed as number (percentage), median (range).

BMI: body mass index, DM: diabetes mellitus, PV: portal vein, CA: celiac axis.

five underwent US-guided drainage, and four underwent EUS-guided drainage for the PFC (Table 2). Several interventions other than PFC drainage were also performed as follows: nine patients underwent EPD, six underwent TAE, and two underwent endoscopic hemostasis for fistula-related post-pancreatectomy hemorrhage (PPH). Of these patients, four underwent such interventions without PFC drainage: EPD (n = 1), TAE (n = 1), EPD and TAE (n = 1), and endoscopic hemostasis (n = 1). Among 50 patients in the B-intervention group with PFCs drainage, 36 had non-infected PFCs and 14 had infected PFCs. Four patients who underwent EUS-guided drainage (non-infected: n = 1, infected: n = 3) were excluded from analysis due to possible bacterial contamination during the procedure. Therefore, we included 35 patients with non-infected PFCs and 11 patients with infected PFCs in our predictive factor analysis.

## Comparison of the perioperative characteristics and outcomes among patients with grade B POPF and those with biochemical leak

To evaluate the clinical impact of our therapeutic strategy on the outcome for POPF after DP, the perioperative characteristics and the outcomes of the patients were compared among the BL, B-antibiotics, and B-intervention groups (Table 3). In terms of perioperative characteristics, the B-antibiotics group had significantly higher WBC and CRP levels on POD 3 to 7 (p = 0.003–0.009) than did the BL group. Patients in the B-intervention group were significantly older (p < 0.001) and had a higher PDAC rate (p = 0.001), lower PNI scores (p = 0.009), and a lower laparoscopic surgery rate (p < 0.001) than did those in the B-antibiotic group. In terms of clinical outcomes, the B-intervention group showed significantly longer hospital stays (41 days vs. 17 days, p < 0.001) and a higher overall complication rate (p < 0.001) than did the B-antibiotics group. The duration of hospitalization in the B-antibiotics group was longer than that in the BL group (17 days vs. 14 days, p = 0.0001). However, there were no significant differences in fistula-related readmission and overall complication rates between the groups.

In order to evaluate the effect of laparoscopic surgery on the outcome, subgroup analysis was performed (S1 and S2 Tables). With regard to preoperative characteristics, patients who underwent laparoscopic surgery had significantly lower DM (p = 0.025), PDAC (p < 0.001), and preoperative therapy (p < 0.001) rates and significantly lower PNI scores (p < 0.001). With regard to intraoperative characteristics, patients who underwent

**Table 2. Management of grade B postoperative pancreatic fistula after distal pancreatectomy.**

| Management | Events, n |
|---|---|
| Antibiotics only | 31 (36.5%) |
| Intervention | 54 (63.5%) |
| persistent drainage only > 3 weeks | 7 (8.2%) |
| Additional drainage for PFCs | 43 (50.5%) |
| CT-guided | 34 |
| US-guided | 5 |
| EUS-guided | 4 |
| EPD | 9 (10.6%) |
| TAE | 6 (7.1%) |
| Endoscopic hemostasis | 2 (2.4%) |

Data are expressed as number (percentage).

PFCs: peripancreatic fluid collections, CT: computed tomography, US: ultrasonography, EUS: endoscopic ultrasonography, EPD: endoscopic pancreatic drainage, TAE: transcatheter arterial embolization.

**Table 3. Comparison of the perioperative characteristics and outcomes among patients with grade B postoperative pancreatic fistula and those with biochemical leak.**

| Characteristics | BL | B-antibiotics | B-intervention | P-value | P-value |
|---|---|---|---|---|---|
| | (n = 30) | (n = 31) | (n = 54) | BL vs B-antibiotics | B-antibiotics vs B-intervention |
| **Preoperative characteristics** | | | | | |
| Sex, male/female | 16/14 | 20/11 | 36/18 | 0.644 | 0.512 |
| Age, years | 68 (10–83) | 54 (28–77) | 68 (20–89) | **0.038** | **<0.001** |
| BMI, kg/m$^2$ | 22.2 (15.9–30.6) | 23.2 (18.9–33.5) | 22.4 (13.6–34.7) | 0.100 | **0.039** |
| DM, n | 6 (20.0%) | 2 (6.5%) | 10 (18.5%) | 0.235 | 0.110 |
| PDAC, n | 4 (13.3%) | 4 (12.9%) | 25 (46.3%) | 0.700 | **0.001** |
| Preoperative therapy, n | 1 (3.3%) | 2 (6.5%) | 15 (27.8%) | 0.500 | **0.015** |
| PNI | 48.4 (36.5–61.4) | 50.3 (39.7–60.1) | 47.7 (32.9–58.2) | 0.287 | **0.009** |
| **Intraoperative characteristics** | | | | | |
| Operation time, min | 310 (140–607) | 287 (132–754) | 347 (193–593) | 0.954 | 0.258 |
| Blood loss, mL | 189 (0–2018) | 363 (3–3520) | 511 (0–2042) | **0.037** | 0.834 |
| Closure method, n | | | | 0.051 | 0.311 |
| Stapler | 15 (50.0%) | 10 (32.3%) | 9 (16.7%) | | |
| Hand-sewn suture | 15 (50.0%) | 21 (67.7%) | 45 (83.3%) | | |
| Laparoscopic surgery, n | 21 (70.0%) | 18 (58.1%) | 9 (16.7%) | 0.440 | **<0.001** |
| without splenectomy, n | 4 (13.3%) | 8 (25.8%) | 4 (7.4%) | 0.289 | **0.023** |
| Simultaneous resection of alimentary tract, n | 0 (0%) | 2 (6.5%) | 8 (14.8%) | 0.368 | 0.215 |
| **Postoperative blood examination** | | | | | |
| WBC on POD 3, /μL | 11,060 (1,540–18,670) | 13,285 (8,460–23,220) | 12,590 (5,870–25,450) | **0.003** | 0.778 |
| CRP on POD3, mg/dl | 13.34 (5.37–31.98) | 17.76 (8.85–27.40) | 15.17 (2.47–22.84) | **0.004** | 0.352 |
| WBC on POD 6–7, /μL | 7 470 (3,800–11,560) | 9 540 (7,740–14,900) | 9,880 (4,990–14,180) | **0.007** | 0.970 |
| CRP on POD 6–7, mg/dL | 3.04 (0.81–8.05) | 7.29 (0.97–19.03) | 5.81 (1.47–21.25) | **0.009** | 0.901 |
| **First CT evaluation of PFCs** | | | | | |
| POD | 6 (2–13) | 6 (3–17) | 6 (2–17) | 0.565 | 0.928 |
| Incidence of PFCs | 17 (56.7%) | 23 (74.2%) | 42 (77.8%) | 0.350 | 0.451 |
| CT value, HU | 19.2 (11.1–36.8) | 17.3 (8.6–26.8) | 16.9 (6.9–33.7) | 0.965 | 0.453 |
| CT volume, mL | 8.7 (0–47.5) | 17.3 (0–82.7) | 23.7 (0–274.5) | 0.195 | 0.390 |
| **Clinical outcome** | | | | | |
| Fistula-related readmission, n | 0 (0%) | 0 (0%) | 6 (11.1%) | 1.000 | 0.054 |
| Length of hospital stay, days | 14 (7–42) | 17 (9–71) | 41 (7–248) | **0.001** | **<0.001** |
| Overall complications CD ≧ 3a, n | 2 (6.7%) | 1 (3.2%) | 54 (100%) | 0.319 | **<0.001** |

Data are expressed as number (percentage), median (range).

POPF: postoperative pancreatic fistula, BL: Biochemical leak, BMI: body mass index, DM: diabetes mellitus, PDAC: pancreatic ductal adenocarcinoma, PNI: prognostic nutritional index, POD: post operative day, WBC: White blood cell count, CRP: C-reactive protein, PFCs: peripancreatic fluid collections, CT: computed tomography, HU: hounsfield units, CD: Clavien–Dindo classification.

laparoscopic surgery had significantly lesser blood loss (p < 0.001), a higher concomitant splenectomy rate (p < 0.001), and a lower rate of simultaneous alimentary tract resection (p < 0.001). Although the pre- and intraoperative characteristics were different between the open and laparoscopic surgery groups, the incidence of CR-POPF was equivalent. However, the laparoscopic surgery group had fewer patients with B-intervention (p < 0.001), fewer overall complications (p < 0.001), and shorter hospital stays (p < 0.001) than did those with open surgery.

## Predictive factors for peripancreatic fluid infection

Perioperative factors were compared between the B-intervention non-infected (n = 35) and infected groups (n = 11) using univariate and multivariate analyses (Table 4). Univariate analysis demonstrated that a higher rate of simultaneous resection of the alimentary tract (p = 0.009), higher CT value of the PFC (p < 0.001), and lower PFC volume (p = 0.025) on drainage day were significantly correlated with PFC infection. In the multivariate analysis,

**Table 4. Predictive factors for peripancreatic fluid infection after distal pancreatectomy.**

| Factor | Univariate analysis | | | Multivariate analysis | | | |
|---|---|---|---|---|---|---|---|
| | B-intervention | B-intervention | P-value | β | Odds ratio | CI | P-value |
| | Non-infected (n = 35) | Infected (n = 11) | | | | | |
| **Preoperative characteristics** | | | | | | | |
| Sex, male/female | 24/11 | 6/5 | 0.307 | | | | |
| Age, years | 68 (41–89) | 64 (20–88) | 0.820 | | | | |
| BMI, kg/m$^2$ | 22.9 (15.4–34.7) | 22.7 (15.8–27.3) | 0.924 | | | | |
| DM, n | 7 (20.0%) | 1 (9.1%) | 0.341 | | | | |
| PDAC, n | 18 (51.4%) | 3 (27.3%) | 0.161 | | | | |
| PNI | 46.7 (32.9–58.2) | 49.2 (34.5–53.0) | 0.188 | | | | |
| **Intraoperative characteristics** | | | | | | | |
| Operation time, min | 356 (201–593) | 343 (260–569) | 0.658 | | | | |
| Blood loss, mL | 507 (0–2042) | 393 (5–1250) | 0.255 | | | | |
| Closure method, n | | | 0.968 | | | | |
| Stapler | 7 (20.0%) | 2 (18.2%) | | | | | |
| Hand-sewn suture | 28 (80.0%) | 9 (81.8%) | | | | | |
| Laparoscopic surgery, n | 7 (20.0%) | 2 (18.2%) | 0.584 | | | | |
| without splenectomy, n | 2 (5.7%) | 2 (18.2%) | 0.168 | | | | |
| Simultaneous resection of alimentary tract, n | 3 (8.6%) | 5 (45.5%) | **0.009** | | | | |
| **Postoperative characteristics** | | | | | | | |
| WBC, /uL | | | | | | | |
| POD 3 | 13,640 (6,750–25,450) | 11,930 (5,870–18,130) | 0.532 | | | | |
| POD 6–7 | 10,135 (4,990–14,180) | 10,560 (8,190–13,440) | 0.350 | | | | |
| Drainage day | 12,280 (4,470–28,980) | 14,110 (6,590–26,120) | 0.308 | | | | |
| CRP, mg/dL | | | | | | | |
| POD3 | 14.90 (5.15–38.18) | 15.95 (9.00–28.35) | 0.314 | | | | |
| POD6-7 | 5.07 (1.47–21.25) | 9.68 (2.84–15.74) | 0.070 | | | | |
| Drainage day | 10.10 (0.92–-32.60) | 13.65 (2.31–28.35) | 0.185 | | | | |
| Body temperature, ˚C | | | | | | | |
| POD3 | 37.2 (36.0–39.3) | 37.3 (36.2–37.9) | 0.730 | | | | |
| POD6-7 | 36.9 (36.5–39.0) | 37.0 (36.5–38.9) | 0.416 | | | | |
| Drainage day | 37.3 (36.3–38.6) | 36.9 (36.4–38.9) | 0.456 | | | | |
| Drain removal, POD | 7 (3–45) | 8 (3–35) | 0.848 | | | | |
| **CT evaluation of PFCs on drainage day** | | | | | | | |
| POD | 13 (5–34) | 17 (3–29) | 0.524 | | | | |
| CT value, HU | 16.1 (6.9–28.0) | 26.3 (12.5–35.9) | <**0.001** | 0.311 | 1.365 | 1.101–1.692 | 0.005 |
| CT volume, ml | 91.7 (6.1–512.4) | 36.2 (10.2–182.0) | **0.025** | | | | |

Data are expressed as number (percentage), median (range).

BMI: body mass index, DM: diabetes mellitus, PDAC: pancreatic ductal adenocarcinoma, PNI: prognostic nutritional index, WBC: white blood count, POD: post operative day, CRP: C-reactive protein, PFC: peripancreatic fluid collection, CT: computed tomography, HU: hounsfeild units, CI: confidence interval.

high CT value was identified as an independent predictive factor of PFC infection (odds ratio: 1.356, 95% confidence interval: 1.101–1.692, p = 0.005).

ROC curve analysis was performed for the B-intervention non-infected versus infected groups to elucidate the optimal cut-off CT value of infected PFCs (Fig 3). The area under the curve (AUC) and the optimal cut-off CT value were 0.884 and 23.2 HU (sensitivity 0.818, specificity 0.943, positive predictive value 0.818, negative predictive value 0.943; p < 0.001), respectively.

## Discussion

The present study revealed two novel findings, which have the potential to improve the postoperative management of grade B POPF after DP. First, although the outcomes in the B-antibiotics group were comparable to those in the BL group, they were much better than those in the B-intervention group. Furthermore, the length of hospitalization was remarkably prolonged when interventions were performed. Second, we detected the CT value of the PFC as a predictor of peripancreatic fluid infection. It may be a useful indicator for PFC drainage.

The clinical burden in the B-antibiotics group was similar to that in the BL group. However, once an interventional treatment was performed, the duration of hospitalization was significantly prolonged. Both B-antibiotics and B-interventions were classified as grade B according to the 2016 ISGPF definition, but their outcomes were quite different. Recently, this issue has been highlighted as a limitation of the 2016 ISGPF definition and a novel stratification system

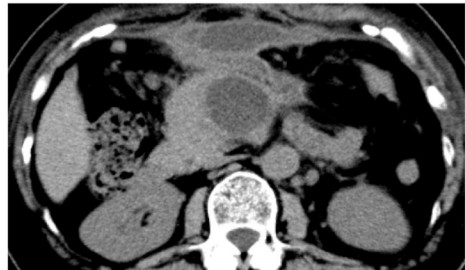

**16.8 HU, infection: (-)**

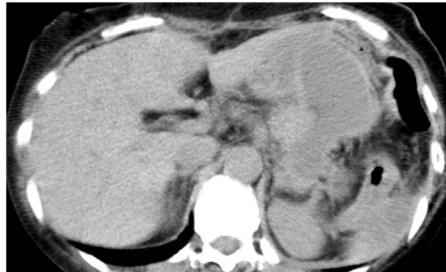

**28.8 HU, infection: (+)** *Pseudomonas aeruginosa*

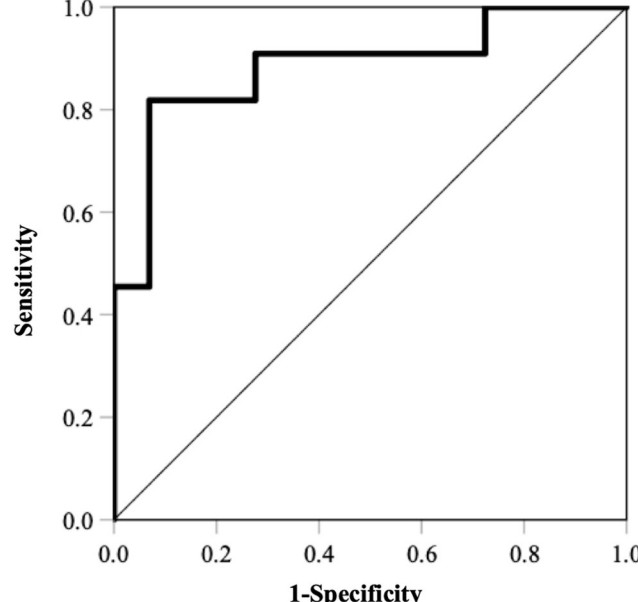

| | ROC analysis | | | | | | |
|---|---|---|---|---|---|---|---|
| | **Cut-off value** | **AUC** | **Sensitivity** | **Specificity** | **PPV** | **NPV** | **P-value** |
| **CT value of infected PFCs** | 23.2 | 0.884 | 0.818 | 0.943 | 0.818 | 0.943 | <0.001 |

**Fig 3. Cut-off value of the CT value of the infected PFCs using ROC curve analysis.** AUC: area under curve, CT: computed tomography, HU: Hounsfield units, ROC: receiver operating characteristic, PFCs: peripancreatic fluid collections, PPV: positive predictive value, NPV: negative predictive value.

had been proposed [12, 21, 22]. Maggino et al. [21] subclassified grade B POPF into three new subcategories (B1, prolonged drainage only; B2, pharmacologic management; B3, interventional procedures) and demonstrated that this stratification permitted improved distinction among different clinical and economic conditions. In this study, patients in the B-intervention group were significantly older and had a higher PDAC rate and lower PNI scores than did those in the B-antibiotics group. These factors were associated with the nutritional status of patients. A previous study revealed that malnutrition was a risk factor for POPF and poor clinical outcomes [23]. If patients with malnutrition developed grade B/C POPF, their illness was more severe, often requiring interventional treatment.

In subgroup analysis to evaluate the effects of laparoscopic surgery, the rates of PDAC and preoperative therapy were significantly higher for patients with open surgery than for those with laparoscopic surgery; this was because we performed laparoscopic surgery only for benign tumors until a few years ago. Despite these background differences, the incidence of CR-POPF was equivalent. However, the clinical outcomes of patients with grade B POPF treated by laparoscopic surgery were much better than those of patients with open surgery. Several reports have revealed that infectious complications are less frequent with laparoscopic surgery than with open surgery [24, 25]. In addition, we assumed that patients with laparoscopic surgery in this study had less frequent or milder intra-abdominal infection because of their higher PNI scores and lower rates of DM and simultaneous alimentary tract resection. These results may have attributed to the lower rate of B-intervention and shorter hospital stays in the laparoscopy group.

In terms of our management of grade B POPF, the rate of additional interventional procedure patients (B3) was higher, and the rates of prolonged drainage-only (B1) and pharmacologic management (B2) patients were lower than those in previous reports (this study: B1/2/3, 8.2%/36.5%/55.3%; previous report: 12.3%–68.0%/40.2%–47.7%/11.8%–40.3%, respectively) [12, 21, 22]. In this study, 53.7% patients in the B-intervention group were diagnosed with non-POPF on POD 3 according to the 2016 ISGPF definition. However, they subsequently developed POPF after removal of intraoperative drain tubes and required additional drainages (Table 5). These results indicated that our method of intraoperative drainage tube replacement (one or two drainage tube in the peripancreatic and/or left subphrenic space) might be inadequate, and leaked pancreatic juices had insufficient drainage, resulting in higher additional drainage rates. Yamashita et al. [5] reported the significance of three prophylactic abdominal drains (to the pancreatic stump, supra-pancreatic space, and left subphrenic space) after DP to minimize the accumulation of non-drained fluid in the abdominal cavity. Moreover, we tended to aggressively perform additional drainage, even in patients with low-grade fever, poor abdominal pain, and mild inflammatory reactions during blood examination. This might be associated with a lower B2 rate and higher B3 rate. Among the patients with B-intervention

**Table 5. Drain amylase level in grade B postoperative pancreatic fistula patients.**

| Drain amylase level | B-antibiotics (n = 31) | B-intervention (n = 54) | P-value |
|---|---|---|---|
| Drain amylase level on POD 3, U/l | 2 234 (405–126 400) | 356 (24–27 475) | < **0.001** |
| > 3 times ULN amylase on POD 3, Yes/ No | 31 (100%) / 0 (%) | 25 (46.3%) / 29 (53.7%) | < **0.001** |
| Drainage day*, POD | - | 13 (3–34) | |
| Drain amylase level on drainage day*, U/L | - | 24,794 (15–183,560) | |

Data are expressed as number (percentage), median (range).

POD: post operative day, ULN: upper limit of normal.

* Four patients without PFC drainage were excluded.

in this study, there might be several patients who could improve with antibiotics alone without overly aggressive drainage and who could be classified into the B-antibiotics group. We should improve our drain management hereafter. In addition, the 2016 ISGPF definition should be updated to be more sensitive and better stratified.

In this study, while the clinical outcomes in the B-antibiotics group were similar to those in the BL group, they were remarkably worse if additional drainage for PFC was performed. As mentioned above, drainage for PFCs tends to be performed aggressively at our hospital. Therefore, in some patients in the B-intervention group, unnecessary drainage may have been performed and required longer hospital stays. To develop a novel approach to decision-making for drainage of PFCs, several factors should be considered. The microbial growth in POPFs was reported to be strongly associated with poor outcomes after pancreatic surgery [13, 14]; therefore, we focused on infection of PFC. Among the 46 patients who underwent additional or persistent drainage, only 11 (23.9%) patients had infected PFCs whereas 35 (76.1%) patients had non-infected PFCs. Absence of PFC infection may indicate that the cause of inflammation is not PFCs but another source. Otherwise, it may reflect that the infection was improving due to antibiotics administered before drainage. Therefore, we postulated that if the development of infected PFCs could be predicted, it would help with the optimal selection of patients with PFCs requiring interventions.

Postoperative inflammatory markers were associated with an increased risk of developing complications after gastrointestinal surgery [26, 27] and clinically relevant pancreatic fistulas after PD [28]. Therefore, we assumed that the postoperative inflammatory markers found among the laboratory data were possible predictors of fluid infection. Surprisingly, there were no significant differences in the inflammatory markers on the day of drainage between the B-intervention infected and non-infected groups. Therefore, the development of an infected PFC cannot be predicted based on inflammatory markers alone. In the univariate analysis for predicting PFC infection, a high rate of simultaneous alimentary tract resection and a low PFC volume were significantly correlated with PFC infection. It could be easily speculated that simultaneous resection of the alimentary tract induced intestinal bacterial contamination. Indeed, microbial growth in POPFs was detected more frequently after PD than after DP [13]. We speculated that infected PFCs tended to be walled-off due to bacterial infection, resulting in lower volume than non-infected PFC. In the multivariate analysis, we observed that the CT value of the PFC was significantly associated with infection at an optimal cut-off value of 23.2 HU. This value was somewhat consistent with that observed in a previous report [15]. Therefore, if a patient developed grade B POPF after DP with elevated inflammatory markers, the CT value of the PFC may be helpful in determining the indications for drainage. Drainage should be performed when the CT value exceeds 23.2 HU, but it may be improved by pharmacological therapy when the value is <23.2 HU. However, the CT value is just one of the predictive factors for PFC infection. Therefore, indications for drainage should be determined according to the assessment of various clinical conditions (e.g., the body temperature, symptoms, the technical challenge of drainage), and optimal drainage should not be delayed.

There were several limitations to this study. First, this was a retrospective, single-center study, with a relatively small sample size. In addition, the study period was long (> 15 years). During this 15-year period, surgical procedures, drain tube replacement, and preoperative management would have developed and the outcomes improved, which could have biased our findings. The indications for additional drainage were different among surgeons, which may have also biased the results. In this study, the incidence of CR-POPF was 34.0% (88/259 cases), which is equivalent to those described in previous single-center studies (37.0–49.1%) [5–7] but is higher than recent multicenter randomized controlled trials (11.0–19.4%) [2–4]. Although the rates of CR-POPF are difficult to compare due to the variations in treatment strategies for

POPF at the different institutions, we should improve our surgical skills and POPF management to achieve better outcomes.

In conclusion, the clinical outcomes of the patients with grade B POPF who received therapeutic antibiotics only were similar to those of the BL patients, but considerably better than those of patients who underwent interventions. To avoid unnecessary drainage and achieve better outcomes, CT values could be useful in predicting PFC infection and determining the indications for drainage in grade B POPF patients with PFCs after DP.

## Supporting information

**S1 Fig. Antibiotic therapy profiles for patients with grade B postoperative pancreatic fistula.** Carbapenem and piperacillin/tazobactam were used frequently in both groups. The duration of antibiotic therapy was significantly longer while the number of used antibiotics regimens was significantly greater in the B-intervention group than in the B-antibiotics group. (TIF)

**S1 Table. Comparison of patient backgrounds and POPF rates between the open and laparoscopic surgery groups.**
(DOCX)

**S2 Table. Comparison of the treatments for grade B POPF between the open and laparoscopic surgery groups.**
(DOCX)

## Acknowledgments

We would like to thank Editage (www.editage.com) for English language editing.

## Author Contributions

**Conceptualization:** Koki Maeda, Naohisa Kuriyama.

**Data curation:** Koki Maeda, Yuki Nakagawa, Takahiro Ito, Yusuke Iizawa, Yasuhiro Murata, Akihiro Tanemura, Masashi Kishiwada, Hiroyuki Sakurai.

**Investigation:** Koki Maeda, Takahiro Ito.

**Methodology:** Naohisa Kuriyama, Takehiro Fujii.

**Resources:** Aoi Hayasaki, Kazuyuki Gyoten, Takehiro Fujii, Yusuke Iizawa, Yasuhiro Murata, Akihiro Tanemura, Masashi Kishiwada, Hiroyuki Sakurai.

**Supervision:** Shugo Mizuno.

**Writing – original draft:** Koki Maeda.

**Writing – review & editing:** Naohisa Kuriyama, Shugo Mizuno.

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
