## [Decision Letter · Decision Letter 0]

9 Aug 2021

PONE-D-21-23713

Optimal management of peripancreatic fluid collection with postoperative pancreatic fistula after distal pancreatectomy: Significance of computed tomography values for predicting fluid infection

PLOS ONE

Dear Dr. Kuriyama,

Thank you for submitting your manuscript to PLOS ONE. After careful consideration, we feel that it has merit but does not fully meet PLOS ONE’s publication criteria as it currently stands. Therefore, we invite you to submit a revised version of the manuscript that addresses the points raised during the review process.

We look forward to receiving your revised manuscript.

Kind regards,

Ulrich Wellner, Prof Dr. med.

Academic Editor

PLOS ONE

Journal Requirements:

a) Did participants provide their written or verbal informed consent to participate in this study?

3. We have noted that this study was designed as a retrospective study. As such, please clarify the nature of the informed consent in your Ethics Statement and Methods section. Did patients consent to the medical treatment, and/or did they specifically consent to participate in this study or to have their medical records used in research? Furthermore, please ensure that you have discussed whether all data/samples were fully anonymized before you accessed them.

Reviewers' comments:

Reviewer's Responses to Questions

**Comments to the Author**

1. Is the manuscript technically sound, and do the data support the conclusions?

Reviewer #1: Yes

Reviewer #2: Partly

2. Has the statistical analysis been performed appropriately and rigorously? 

Reviewer #1: Yes

Reviewer #2: Yes

3. Have the authors made all data underlying the findings in their manuscript fully available?

Reviewer #1: Yes

Reviewer #2: No

4. Is the manuscript presented in an intelligible fashion and written in standard English?

Reviewer #1: Yes

Reviewer #2: No

5. Review Comments to the Author

Reviewer #1: Dear authors and editors:

Manuscript titled" Optimal management of peripancreatic fluid collection with postoperative pancreatic fistula after distal pancreatectomy: Significance of computed tomography values for predicting fluid infection" is a research about how to evaluate the management and outcomes of patients with postoperative pancreatic fistula(POPF) and verify the significance of computed tomography values for predicting peripancreatic fluid infections after distal pancreatectomy.

Peripancreatic fluid collections(PFC) are very common clinical manifestations after distal pancreatectomy. However, optimal management remains unclear. Of these PFC, diagnosed as whether or not PF or infection, the authors figured out a best way to evaluate、predict and deal with them respectively. Well done！

Reviewer #2: This research extends our understanding of pancreatic fistula and the treatment options. There are 3 main problems: 1. They initiate antibiotics in both 2 groups, which means the time and the regimens of antibiotics are essential to deliver a solid conclusion. But in the manuscript, the author failed to give such data. 2.The 2 cohort showed significant difference in age and pancreatic cancer rate. As we all know, age and malignancy are vital factors for pancreatic fistula prevalence. Thirdly, the author should give a sub group analysis to compare the effcets between open surgery and laparoscopic surgery.

6. PLOS authors have the option to publish the peer review history of their article (what does this mean?). If published, this will include your full peer review and any attached files.

Reviewer #1: No

Reviewer #2: No

---

## [Author Response · Author response to Decision Letter 0]

31 Aug 2021

Journal Requirements:

→We have ensured that our manuscript continues to adhere to the requirements of PLOS ONE.

a) Did participants provide their written or verbal informed consent to participate in this study?

→ We have amended our ethics statement in accordance with your instructions. We could not obtain written informed consent from the participants because of the retrospective nature of the study. Informed consent was obtained through an opt-out form approved by our ethics committee.

3. We have noted that this study was designed as a retrospective study. As such, please clarify the nature of the informed consent in your Ethics Statement and Methods section. Did patients consent to the medical treatment, and/or did they specifically consent to participate in this study or to have their medical records used in research? Furthermore, please ensure that you have discussed whether all data/samples were fully anonymized before you accessed them.

→ We could not obtain written informed consent from the participants because of the retrospective nature of the study. Informed consent was obtained through an opt-out form approved by our ethics committee. All data were fully anonymized before we accessed them. These procedures were approved by our ethics committee. Accordingly, we have revised the text as shown below. 

P4, line 8: 

The protocol for this research was approved by a suitably constituted Ethics Committee at the institution (Committee of the Institutional Review Board at Mie University of Japan, Approval No. H2021-024), and the study conformed to the provisions of the Declaration of Helsinki. Prior to their inclusion in this retrospective study, individual participants provided informed consent for participation and the use of their medical records through an opt-out form. This consent procedure was approved by the ethics committee. All data were fully anonymized before we accessed them.

Reviewer #1: Dear authors and editors:

Manuscript titled" Optimal management of peripancreatic fluid collection with postoperative pancreatic fistula after distal pancreatectomy: Significance of computed tomography values for predicting fluid infection" is a research about how to evaluate the management and outcomes of patients with postoperative pancreatic fistula(POPF) and verify the significance of computed tomography values for predicting peripancreatic fluid infections after distal pancreatectomy.

Peripancreatic fluid collections(PFC) are very common clinical manifestations after distal pancreatectomy. However, optimal management remains unclear. Of these PFC, diagnosed as whether or not PF or infection, the authors figured out a best way to evaluate、predict and deal with them respectively. Well done！

→ Thank you for your positive feedback and appreciation.

Reviewer #2: This research extends our understanding of pancreatic fistula and the treatment options. There are 3 main problems: 

1. They initiate antibiotics in both 2 groups, which means the time and the regimens of antibiotics are essential to deliver a solid conclusion. But in the manuscript, the author failed to give such data.

→We apologize for the inadequate details regarding the antibiotic regimens. We have now provided this information in a supplementary figure (S1 Fig). We have also added the relevant information in the manuscript, as shown below.

Results P8, line 4:

The duration and regimens of antibiotic therapy in both groups are shown in S1 Fig.

P23, line 2:

S1 Fig. Antibiotics therapy profiles for patients with grade B postoperative pancreatic fistula

Carbapenem and piperacillin/tazobactam were used frequently in both groups. The duration of antibiotic therapy was significantly longer, while the number of used antibiotics regimens was significantly greater, in the B-intervention group than in the B-antibiotic group.

2.The 2 cohort showed significant difference in age and pancreatic cancer rate. As we all know, age and malignancy are vital factors for pancreatic fistula prevalence. 

→Thank you for pointing this out. As you mentioned, patients in the B-intervention group were significantly older and had a higher PDAC rate than did those in the B-antibiotics group in this study. We have added the relevant text in the revised manuscript, as shown below.

Discussion P14, line 16

In this study, patients in the B-intervention group were significantly older and had a higher PDAC rate and lower PNI scores than did those in the B-antibiotics group. These factors were associated with the nutritional status of patients. A previous study revealed that malnutrition was a risk factor for POPF and poor clinical outcomes [23]. If patients with malnutrition developed grade B/C POPF, their illness was more severe, often requiring interventional treatment.

REFERENCE

23. Kim E, Kang JS, Han Y, Kim H, Kwon W, Kim JR, et al. Influence of preoperative nutritional status on clinical outcomes after pancreatoduodenectomy. HPB. 2018;20: 1051–61.

Thirdly, the author should give a sub group analysis to compare the effcets between open surgery and laparoscopic surgery.

→Thank you for your excellent suggestions. We performed a subgroup analysis of the two surgical procedures. However, the patient backgrounds were significantly different between the two groups. Because we performed laparoscopic surgery only for benign tumors until a few years ago, the rates of PDAC and preoperative therapy were significantly higher in the open group than in the laparoscopy group. Therefore, we decided to add the results of subgroup analysis in supplementary tables. We have also added the relevant text in the Results and Discussion sections.

Result P10, line 15

In order to evaluate the effect of laparoscopic surgery on the outcome, subgroup analysis was performed (S1 and S2 Tables). With regard to preoperative characteristics, patients who underwent laparoscopic surgery had significantly lower DM (p = 0.025), PDAC (p < 0.001), and preoperative therapy (p < 0.001) rates and significantly lower PNI scores (p < 0.001). With regard to intraoperative characteristics, patients who underwent laparoscopic surgery had significantly lesser blood loss (p < 0.001), a higher concomitant splenectomy rate (p < 0.001), and a lower rate of simultaneous alimentary tract resection (p < 0.001). Although the pre- and intraoperative characteristics were different between the open and laparoscopic surgery groups, the incidence of CR-POPF was equivalent. However, the laparoscopic surgery group had fewer patients with B-intervention (p < 0.001), fewer overall complications (p < 0.001), and shorter hospital stays (p < 0.001) than did those with open surgery.

Discussion P15, line 3

In subgroup analysis to evaluate the effects of laparoscopic surgery, the rates of PDAC and preoperative therapy were significantly higher for patients with open surgery than for those with laparoscopic surgery; this was because we performed laparoscopic surgery only for benign tumors until a few years ago. Despite these background differences, the incidence of CR-POPF was equivalent. However, the clinical outcomes of patients with grade B POPF treated by laparoscopic surgery were much better than those of patients with open surgery. Several reports have revealed that infectious complications are less frequent with laparoscopic surgery than with open surgery [24,25]. In addition, we assumed that patients with laparoscopic surgery in this study had less frequent or milder intra-abdominal infection because of their higher PNI scores and lower rates of DM and simultaneous alimentary tract resection. These results may have attributed to the lower rate of B-intervention and shorter hospital stays in the laparoscopy group. 

Reference

24. Targarona EM, Balague C, Knook MM, Trias M. Laparoscopic surgery and surgical infection. Br J Surg. 2000;87: 536–44. 

25. Venkat R, Edil BH, Schulick RD, Lidor AO, Makary MA, Wolfgang CL. Laparoscopic distal pancreatectomy is associated with significantly less overall morbidity compared to the open technique: A systematic review and meta-analysis. Ann Surg. 2012;255: 1048–59. 

P23, line 7 and 10

S1 Table. Comparison of patient backgrounds and POPF rates between the open and laparoscopic surgery groups.

S2 Table. Comparison of the treatments for grade B POPF between the open and laparoscopic surgery groups.

---

## [Editor Report · Decision Letter 1]

25 Oct 2021

Optimal management of peripancreatic fluid collection with postoperative pancreatic fistula after distal pancreatectomy: Significance of computed tomography values for predicting fluid infection

PONE-D-21-23713R1

Dear Dr. Kuriyama,

We’re pleased to inform you that your manuscript has been judged scientifically suitable for publication and will be formally accepted for publication once it meets all outstanding technical requirements.

Kind regards,

Ulrich Wellner, Prof Dr. med.

Academic Editor

PLOS ONE
---

## [Editor Report · Acceptance letter]

28 Oct 2021

PONE-D-21-23713R1 

Optimal management of peripancreatic fluid collection with postoperative pancreatic fistula after distal pancreatectomy: Significance of computed tomography values for predicting fluid infection 

Dear Dr. Kuriyama:

I'm pleased to inform you that your manuscript has been deemed suitable for publication in PLOS ONE. Congratulations! Your manuscript is now with our production department. 

Kind regards, 

on behalf of

Mr. Ulrich Wellner 

Academic Editor

PLOS ONE